# Phospho-DIGE Identified Phosphoproteins Involved in Pathways Related to Tumour Growth in Endometrial Cancer

**DOI:** 10.3390/ijms241511987

**Published:** 2023-07-26

**Authors:** Valeria Capaci, Giorgio Arrigoni, Lorenzo Monasta, Michelangelo Aloisio, Giulia Rocca, Giovanni Di Lorenzo, Danilo Licastro, Federico Romano, Giuseppe Ricci, Blendi Ura

**Affiliations:** 1Institute for Maternal and Child Health, IRCCS Burlo Garofolo, 34137 Trieste, Italy; valeria.capaci@burlo.trieste.it (V.C.); michelangelo.aloisio@burlo.trieste.it (M.A.); giovanni.dilorenzo@burlo.trieste.it (G.D.L.); federico.romano@burlo.trieste.it (F.R.); giuseppe.ricci@burlo.trieste.it (G.R.); blendi.ura@burlo.trieste.it (B.U.); 2Department of Biomedical Sciences, University of Padova, 35131 Padova, Italy; giorgio.arrigoni@unipd.it (G.A.); giulia.rocca@unipd.it (G.R.); 3Proteomics Center, University of Padova and Azienda Ospedaliera di Padova, 35131 Padova, Italy; 4CRIBI Biotechnology Center, University of Padova, 35131 Padova, Italy; 5AREA Science Park, Basovizza, 34149 Trieste, Italy; licastrod@gmail.com; 6Department of Medicine, Surgery and Health Sciences, University of Trieste, 34129 Trieste, Italy

**Keywords:** phosphoproteins, 2D-DIGE, mass spectrometry, endometrial cancer

## Abstract

Endometrial cancer (EC) is the most common gynecologic malignancy of the endometrium. This study focuses on EC and normal endometrium phosphoproteome to identify differentially phosphorylated proteins involved in tumorigenic signalling pathways which induce cancer growth. We obtained tissue samples from 8 types I EC at tumour stage 1 and 8 normal endometria. We analyzed the phosphoproteome by two-dimensional differential gel electrophoresis (2D-DIGE), combined with immobilized metal affinity chromatography (IMAC) and mass spectrometry for protein and phosphopeptide identification. Quantities of 34 phosphoproteins enriched by the IMAC approach were significantly different in the EC compared to the endometrium. Validation using Western blotting analysis on 13 patients with type I EC at tumour stage 1 and 13 endometria samples confirmed the altered abundance of HBB, CKB, LDHB, and HSPB1. Three EC samples were used for in-depth identification of phosphoproteins by LC-MS/MS analysis. Bioinformatic analysis revealed several tumorigenic signalling pathways. Our study highlights the involvement of the phosphoproteome in EC tumour growth. Further studies are needed to understand the role of phosphorylation in EC. Our data shed light on mechanisms that still need to be ascertained but could open the path to a new class of drugs that could hinder EC growth.

## 1. Introduction

Endometrial cancer (EC) is one of the most common gynecologic malignancies, affecting 3% of women [1]. Annually 10–20 per 100,000 women develop this type of cancer, and its incidence and mortality continuously increase [2]. ECs arise primarily from the endometrium’s glandular tissue, the uterus’s inner part. Clinically, two types of EC can be distinguished: estrogen-dependent type I and estrogen-independent type II [3]. Type I ECs represent around 90% of all ECs, frequently display endometrioid cell morphology, low grade (I or II) and are associated with a good prognosis. Conversely, type II is rarer, characterized by serous clear cells, undifferentiated carcinomas or carcinosarcomas, and shows poor prognosis [4]. Most ECs occur as sporadic tumours, being health and lifestyle conditions including obesity, metabolic syndrome, diabetes, polycystic ovary syndrome and high estrogen levels relevant to EC-predisposing factors [5,6,7,8,9,10].

Mechanisms underlying the carcinogenesis of EC are poorly known. In the last 20 years, several regulatory processes involved in ECs pathomechanism have been identified, being the disruptions of Wnt/β-catenin [11], Akt/PI3K/mTOR [12,13,14], MAPK/ERK [15] and VEGF/VEGFR [16] pathways, together with mutations in TP53-binding protein 1 (TP53) and Cyclin-dependent kinase inhibitor 2A (P16), the most common found in ECs or EC cell lines [17,18]. In particular, type I frequently carries a mutation in phosphatidylinositol 3-kinase (PI3K), Catenin beta-1 (CTNNB1), GTPase KRas (KRAS), and POLE (encoding for one domain of polymerase- ε) genes. In contrast, most type II tumours carry gain of function mutations in the TP53 gene and, at lower extent, mutation on (F-box/WD repeat-containing protein 7) FBXW7, (AT-rich interactive domain-containing protein) ARID1A or (Receptor tyrosine-protein kinase erbB-2) ERBB2 gene amplification [4,19]. Genetic mutations and epigenetic changes lead to the deregulation of signal transduction pathways causing changes in standard cellular mechanisms and promoting tumour development and aggressiveness [20]. Phosphorylation/dephosphorylation events mediate cellular transduction signalling due to specific kinases and phosphatases. Indeed, protein phosphorylation is an essential control mechanism of cellular processes, including proliferation, cell death, metabolism, and intracellular trafficking [21]. Derangement of protein phosphorylation, including hyperactivation, malfunction or overexpression of kinases, is found in cancer.

Therefore, a more profound knowledge of the role of these signalling transduction pathways and the cross-talk between them is necessary to understand how ECs originate and identify new targeted therapy drugs. Mass spectrometry is a compelling technology for identifying cancer biomarkers [22,23] and understanding tumour pathophysiology mechanisms [24]. Phosphoproteomics has critical relevance for comprehending the molecular mechanisms involved in cancer, and the identification of phospho-proteome in EC might open a new therapeutic window for its treatment. Yongchao Dou et al. conducted a phospho-protein study on many patients giving information about proteomic markers involved in pathways leading to tumour development [19]. A similar successful approach led to identifying specific, actionable targets and the introduction of particular drugs for several types of cancers in clinics. [25,26]. Similarly, the identification of aberrant signalling downstream of (Epidermal growth factor receptor) EGFR mutation and (Receptor tyrosine-protein kinase erbB-2) Her2 amplification opened new therapeutic windows in lung, breast and colon cancer [27,28].

Indeed, in this work, we performed an analysis of phosphoproteome by using phosphor 2D-DIGE and MS in patients at an early stage of type I EC, with particular attention to the importance of phosphorylation of proteins involved in crucial signalling pathways, to identify processes that contribute to the onset and progression of EC.

## 2. Results

### 2.1. Proteomic Study

We used phosphor-DIGE coupled with MS of 8 EC and 8 CTRL for phosphoproteomics analysis. Proteomweaver software detected more than 2000 spots in both types of samples (Figure 1).

We identified 34 proteins that revealed a significant alteration Mann–Whitney sum-rank test (*p* < 0.05) (Table 1). Our data proved that 31 were increased in EC with fold change ≥ 1.5 and 3 proteins decreased in EC with fold change ≤ 0.6 (Appendix A). Of them, 23 had been identified as phosphoproteins in EC by Yongchao Dou and colleagues (TXN, ARPC5, HBB, HSPB6, HSPB1, PSMA3, EEF1D, P4HB, CKB, PDIA6, GPI, HSPA5, ERP29, CAPZB, ANXA3, LDHB, AHCY, SNX6, WARS1, AHCY, ENO1, STIP1, HNRNPD). We conducted an LC-MS/MS analysis on 3 type I EC tissue samples at stage 1. We identified 552 phosphoproteins and 942 phosphopeptides by LC-MS/MS analysis (Appendix A), among which 12 proteins (TXN, HBB, HSPB1, PSMA3, EEF1D, P4HB, PDIA6, HSPA5, ACTG2, ENO1, H4C1, HNRNPD) were identified in 2D-DIGE experiments, thus directly confirming their phosphorylation status. Two phosphoproteins were recognised only in our analysis (ACTG2, H4C1). In total, 26 proteins determined by 2D-DIGE were confirmed as phosphoproteins from both studies, while in the Phosphositeplus database, all the specified proteins were reported to be phosphorylated.

### 2.2. Western Blotting

For the validation of the phosphoproteomic data, we selected three up-regulated phosphoproteins with fold change ≥ 4.5 (HBB, HPB1, LDHB) and one downregulated protein (CKB) with fold change ≤ 0.28 (Figure 2) (Appendix A). We used western blotting to validate the abundance of the phosphoproteins in 13 ECs vs. 13 CTRLs. A higher abundance was significant (Mann–Whitney sum-rank test *p* < 0.05) for HBB (*p* = 0.045) and HSPB1 (*p* = 0.0096), while LDHB showed an increase but not statistically noteworthy (*p* = 0.75). A decreased abundance was meaningful for CKB (*p* = 0.0019).

### 2.3. Bioinformatic Analysis

To understand the functional impact of differentially phosphorylated protein between ECs versus normal endometrium, we next performed bioinformatic enrichment using the gProfiler classification tool to categorize proteins based on their molecular function, biological processes, and cellular component (Figure 3). In terms of molecular function, proteins were ranked into intramolecular oxidoreductase activity, protein folding chaperone, protein dimerization activity, protein homodimerization activity, unfolded protein binding, protein-disulfide reductase activity, disulfide oxidoreductase activity, and haemoglobin binding. Regarding the biological processes, proteins were categorized into protein folding, cell death regulation, cell death, protein metabolic process regulation, protein refolding, catalytic activity negative regulation, catalytic activity regulation, and mesenchyme migration. Considering the cellular compartment, proteins were organized into extracellular exosome, extracellular vesicle, extracellular organelle, extracellular membrane-bounded organelle, vesicle, extracellular space, extracellular region, and cytosol. Regarding the pathway, Reactome’s tools indicate that these proteins are primarily involved in scavenging heme from plasma, Neutrophil degranulation, RNA Polymerase II Transcription Termination and AUF1 (hnRNP D0) binds and destabilizes mRNA.

IPA analysis (Figure 4) showed that identified proteins are required in top networks corresponding to (1) Post-translational modification and protein folding (ERP29, HSPA5, HSPB1, PDIA6, PFDN2, TXN) (2) Cell Viability (CTSB, ENO1, HBB, HSPA5, HSPB1, HSPB6, MAGOH, P4HB, PFDN2, PSMA3, STIP1, TXN) (3) Free Radical Scavenging, Small Molecule Biochemistry (HBB, TXN) (4) Cellular Infiltration (ANXA3, CTSB, ENO1, HSPA5, HSPB1, P4HB).

## 3. Discussion

PTMs (post-translational modifications) are biochemical events that occur after translation and can affect the proteins’ conformation, activity, and location in the cell [29]. There are more than 300 types of PTMs, and the most important of these, phosphorylation, is a modification related to carcinogenesis and cancer development, regulating numerous cellular processing, including cell growth, proliferation, death, metabolism, and signal transduction [30]. Mechanisms underlying the carcinogenesis of EC are poorly known. Therefore, a more profound knowledge of EC phosphoproteomics might have critical relevance for understanding molecular mechanisms and signalling transduction pathways underlying ECs development.

For this reason, we decided to conduct a phosphoproteomics study employing IMAC enrichment, 2D-DIGE and MS analysis identifying 38 phosphoproteins differently abundant in the EC tissue compared with the endometrial tissue. In addition, utilizing western blotting analysis, we confirmed in a new cohort of samples the significantly altered abundance of 3 phosphoproteins—HBB, CKB, HSPB1—in the EC vs. normal endometrial tissue.

HSPB1 is a small heat shock protein involved in environmental stress response that, upon activation, translocates from the cytoplasm to the nucleus and functions as a molecular chaperone. In response to several signals, it is phosphorylated by different kinases, including p90Rsk (ribosomal s6 kinase), PKC (Protein kinase C), PKD (Protein kinase D), PKG (Protein kinase G) and MAPKAP (MAP kinase-activated protein) kinases 2 and 3 [31]. The exact mechanism by which HSPB1 regulates cancer growth is still unknown [32]; however, this small protein is involved in several tumours and takes part in several distinctive cancer features, including resistance to stress response and actin organization [33]. Interestingly, data reported in the literature indicate that the phosphorylation and dephosphorylation state of HSPB1 controls the apoptosis of cancer cells [34,35]. Phosphorylation of HSPB1 in S15 and S82 is the most up-regulated event in treatment with chemotherapy in pancreatic cancer, raising the tumour resistance. Inhibition of this protein leads to increased apoptosis by chemotherapy [36].

LDHB, although not significantly increased, is a glycolytic enzyme that converts pyruvate and lactate [37]. This enzyme regulates autophagy and lysosomal activity leading to mainly lactate rather than glucose [38]. LDHB is phosphorylated primarily by the Aurora-A kinase enzyme. [39]. Aurora-A phosphorylation on Ser 162 induces the augmented glycolytic activity of LDHB, leading to tumour growth [40]. The two phosphorylated sites, S85 and S90, reported in this study were identified in non-small cell lung cancer [41]. CKB encodes for an enzyme that catalyzes the transfer reaction of phosphate between ATP and creatine phosphate, controlling energy homeostasis [42]. Its role in cancer, and especially in ECs, is obscure. Xu-Hui Li and colleagues have provided new evidence on the association between the expression of this gene and the survival and progression of ovarian cancer cells [43]. Phosphorylation of T35 is involved in different types of breast cancer, including HER2-positive breast cancer, luminal A and B breast cancer [44].

Considering HBB enrichment, its function is of great interest. Recently Yu Zheng et al. performing a single-cell RNA-Seq study on different types of tumours, found that HBB might induce a cytoprotective effect against the metastasis process [45]. Another study highlighted the cytoprotective role of the protein from oxidative stress in cervical cancer [46]. HBB protein phosphorylation in S10, S45, and T124 has been identified in breast cancer cells [41], while the phosphorylation in T5 has been detected in hepatocellular carcinoma [47] and pancreatic ductal adenocarcinoma [48].

Next, we performed bioinformatic analysis to understand the functional impact of differentially phosphorylated proteins between ECs compared to normal endometrium. Subsequently, we performed bioinformatic analysis to understand the functional impact of differentially phosphorylated proteins between ECs compared to normal endometrium. Our Reactome analysis revealed the possible involvement of the Neutrophil degranulation pathway in the early stage of EC. Neutrophils are the first cell responding to stimuli in infection and inflammation. They interact with cancer cells, allowing them to bind to the endothelium and promoting tumour advancement. Neutrophils can induce tumour progression through different mechanisms such as matrix remodelling, angiogenesis stimulation, and immunosuppression. An inflammatory response mediated by neutrophils can induce the activation of dormant cancer cells and a reactivation of the tumour. Their degranulation leads to the release of soluble granules with membrane proteins into cell surfaces, making possible extravasation of circulating tumour cells and their metastasization [49].

A limitation of this study is the limited number of phosphoproteins. A global LC/MS-MS phosphoprotein study would provide more information on possible pathological mechanisms of EC. Another limitation of this study is the need for more functional studies, which are required to ascertain the involvement of this phosphoprotein in EC development. We are organizing research in established/PDX endometrial cancer cells, transplanting them in immunocompromised mice to study the mechanisms of these key proteins.

In conclusion, proteomics highlighted the involvement of phosphoproteins in the early stages of EC. Bioinformatics tools have revealed several pathways related to oxidative stress, tumour development, and phosphoproteins involved in cell viability, cancer infiltration, and protein folding. Other studies are needed to understand the involvement of phosphorylation in the early stages of tumour growth.

## 4. Materials and Methods

### 4.1. Patients

During 2021 and 2022, a total of 26 patients (13 women suffering from EC and 13 non-EC controls) were recruited at the Institute for Maternal and Child Health—IRCCS “Burlo Garofolo” (Trieste, Italy). All EC patients had a type I EC at stage I. All procedures complied with the Declaration of Helsinki and were approved by the Institute’s Technical and Scientific Committee. All patients signed informed consent forms. In Appendix A, we describe the clinical and pathological characteristics of the patients. The median age of patients was 70 years (IQR 59–74, Min = 59, Max = 74). The median age of controls was 44 years (IQR 42–47, Min = 32, Max = 51). The endometrial tissue samples were derived from ordinary leiomyomas in the selection of the controls. We excluded patients and controls with Human immunodeficiency virus (HIV), Hepatitis B virus (HBV), Hepatitis C virus (HCV), leiomyoma and adenomyosis.

### 4.2. Phosphoprotein and Phosphopeptide Isolation

The phosphoprotein isolation from EC and healthy endometrium tissue samples was performed by Phosphoprotein Enrichment Kit (Pierce, Appleton, USA) [50]. About 5 mg of tissue from the EC or the healthy endometrium was homogenized in lysis buffer (1% NP-40, 50 mM Tris-HCl (pH 8.0), NaCl 150 mM) with Phosphatase Inhibitor Cocktail Set II 1× (Millipore, Burlington, VT, USA) and 2 mM phenylmethylsulfonyl fluoride (PMSF), 1 mM benzamidine. Protein concentration was determined by Bradford assay. According to the manufacturer’s instructions, 3mg of tissue lysate was used for the phosphoprotein purification by immobilized metal affinity chromatography (IMAC).

3 EC was used to isolate phosphopeptides. Tissues were lysed, and proteins were digested with EasyPep Maxi MS Sample Prep Kit (Thermo Fisher, Boston, MA, USA). After protein digestion, peptides were transferred in dioxide (TiO_2_) spin tips (Thermo Fisher) for phosphopeptide enrichment, according to the manufacturer’s instructions.

### 4.3. Sample Preparation for 2D-DIGE and Gel Image Analysis

For 2D-DIGE experiments, 50 µg of phosphoproteins-enriched samples from EC patients and controls were labelled with 400 pmol of either Cy5 or Cy3. An equal sample quantity was pooled and labelled with Cy2 for the internal standard preparation. The reaction labelling was stopped by adding 1 µL of 10 mM lysine. After labelling, proteins were diluted in a rehydration buffer: 7 M urea, 2 M thiourea, 2% (*w*/*v*) CHAPS, 65 mM DTT, and 0.24% Bio-Lyte [3,4,5,6,7,8,9,10] (Bio-Rad Laboratories, Inc., Hercules, CA, USA). The 2-DE analysis was conducted as previously described [51]. ReadyStrip™ 4–7 18 cm immobilized pH gradient (IPG) (Bio-Rad Laboratories, Inc., Hercules, CA, USA) strips were used for 2-DE. Strips were rehydrated at 50 V for 12 h at 20 °C, and isoelectric focusing (IEF) was performed in a PROTEAN IEF Cell (Bio-Rad Laboratories, Inc., Hercules, CA, USA). Strips equilibration was achieved with two incubations: the first equilibration in a buffer (6 M urea, 2% SDS, 50 mM Tris-HCl (pH 8.8), 30% glycerol) for 5 min and a second equilibration step performed in 4% iodoacetamide for 10 min. The IPG strips were transferred to a 12% polyacrylamide gel (18.5 cm × 20 cm). Protein molecular weights were determined by Precision Plus Protein Prestained Standards (Bio-Rad Laboratories, Inc., Hercules, CA, USA) with a molecular weight range from 10 to 250 kDa. Gel analysis was executed by Proteomweaver 4.0 software (both from Bio-Rad Laboratories, Inc., Hercules, CA, USA) to normalize and quantify protein spots. Two experimental replicates per sample were performed. Polyacrylamide gel was scanned with a Molecular Imager PharosFX System (Bio-Rad Laboratories, Inc., Hercules, CA, USA).

### 4.4. Western Blotting

Western blotting analysis confirmed the altered abundance of 4 phosphoproteins, namely HBB, HPB1, LDHB and CKB, using the IMAC-enriched proteins derived from 13 ECs and 13 healthy samples, as previously described [52]. The phosphorylation status of these proteins was directly verified by LC-MS/MS (as described below) or by retrieving the information from the study by Yongchao Dou et al. [19]. For this analysis, 20 µg of phosphoproteins were loaded on 4–20% precast gel and then transferred to a nitrocellulose membrane. Once the proteins were transferred, the membrane was blocked by treatment with 5% defatted milk in TBS-tween 20 and incubated overnight at 4 °C with 1:500 diluted primary rabbit polyclonal antibody against HBB, with 1:700 diluted primary rabbit polyclonal antibody against HPB1, with 1:1000 diluted primary rabbit polyclonal antibody against LDHB, 1:1000 diluted primary rabbit polyclonal antibody against CKB. After primary antibody incubation, the membranes were washed with TBS-Tween 0.05% and incubated with HRP-conjugated anti-rabbit IgG and anti-mouse IgG (1:3000, Sigma-Aldrich; Merck KGaA, Darmstadt, Germany). SuperSignal West Pico Chemiluminescent (Thermo Fisher Scientific Inc., Ottawa, ON, Canada) was used for protein band signal visualization. The intensities of the Immunostained bands were normalized with the total protein intensities measured by staining the membranes from the same blot with Red Ponceau solution (Sigma-Aldrich, St. Louis, MO, USA).

### 4.5. Trypsin Digestion and MS Analysis

For protein spot digestion, a 2-DE gel (200 µg of loaded phosphoproteins) was run and stained with SYPRO Ruby and Coomassie colloidal Blue for protein visualization. Protein spots were selected, digested, and analyzed by mass spectrometry by Ura et al. [53]. Protein spots were washed four times with 50 mM NH_4_HCO_3_ and acetonitrile (ACN; Sigma-Aldrich, St. Louis, MO, USA) and dried under a vacuum in a SpeedVac system. Three microliters of 12.5 ng/µL sequencing grade modified trypsin (Promega, Madison, WI, USA) in 50 mM NH_4_HCO_3_ was used to digest spots. Digestion was conducted overnight at 37^o^ C. Peptide extraction was performed with three changes extraction by 50% ACN/0.1% formic acid (FA; Fluka, Ammerbuch, Germany).

LC-MS/MS analyzed four microliters of each peptide mixture with an LTQ-Orbitrap XL mass spectrometer (Thermo Fisher Scientific, Waltham, MA, USA) coupled to a nano-HPLC Ultimate 3000 (Dionex—Thermo Fisher Scientific). Peptides were separated at 250 nL/min in a 10 cm pico-frit column (75 μm ID, 15 μm Tip; (CoAnn Technologies 350 Hills St, Richland, WA 99354, United States) packed in-house with C18 material (Aeris Peptide 3.6 µm XB-C18, (Phenomenex, Torrance, California, United States) utilizing a linear gradient of ACN/0.1% formic acid from 3% to 40% in 20 min. Data were acquired with a Data Dependent Acquisition (DDA) approach: a full scan in the 300–1700 m/z range was performed at high resolution (60,000) on the Orbitrap. The ten most intense ions in the linear ion trap were selected for CID fragmentation and MS/MS data acquisition in low resolution (Top10 method). Raw data were analyzed with Proteome Discoverer 1.4 (Thermo Fisher Scientific) connected to a Mascot Search Engine (version 2.2.4, Matrix Science, London, UK). Spectra were searched against the human section of the Uniprot database (version 20200901) using the following parameters: enzyme specificity was set on trypsin with one missed cleavage allowed, and the precursor and fragment ions tolerance were 10 ppm and 0.6 Da, respectively. Carbamidomethylation of cys and oxidation of met residues were set as fixed and variable modifications, respectively. The Percolator algorithm assessed the False Discovery Rate (FDR) at protein and peptide levels. We considered those proteins identified with at least three unique peptides and FDR < 0.01 as positive hits. The precursor area detection node of Proteome Discoverer was used to estimate the abundance of each protein in the cases in which multiple proteins were identified within the same spot.

TiO_2_ enriched samples were dried under vacuum and suspended in 50 μL of 3%ACN/0.1% formic acid. 5 μL of each sample was analyzed as described above but using a linear gradient of ACN/0.1% formic acid from 3% to 40% in 40 min. to identify phosphopeptides. The same sample was analyzed three times using the same chromatographic conditions but different instrumental methods: in particular, after the first acquisition with a Top10 DDA method, a second acquisition was performed using a static excluding list containing all peptides successfully identified in the first analysis. The third analysis was performed with a Top3 multi-stage activation method (MSA), described in Venerando et al. [54]. Data analysis was performed as reported above, adding phosphorylation of Ser and Thr residues as variable modifications.

### 4.6. Bioinformatic Analysis

Proteins identified by MS were enriched using the g-Profiler tool for their involvement in the biological processes, molecular function, and protein class. Ingenuity Pathway Analysis (IPA) generated bio-functions with *p* < 0.01. We only considered associations for the filter summary with high confidence or that had been observed experimentally.

### 4.7. Statistical Analysis

Differences were considered significant when spots showed a fold change ± 1.5 and satisfied the Mann–Whitney sum-rank test (*p* < 0.05). All analyses were conducted with Stata/IC 16.1 for Windows (StataCorp LP, College Station, TX, USA).

## Figures and Tables

**Figure 1 ijms-24-11987-f001:**
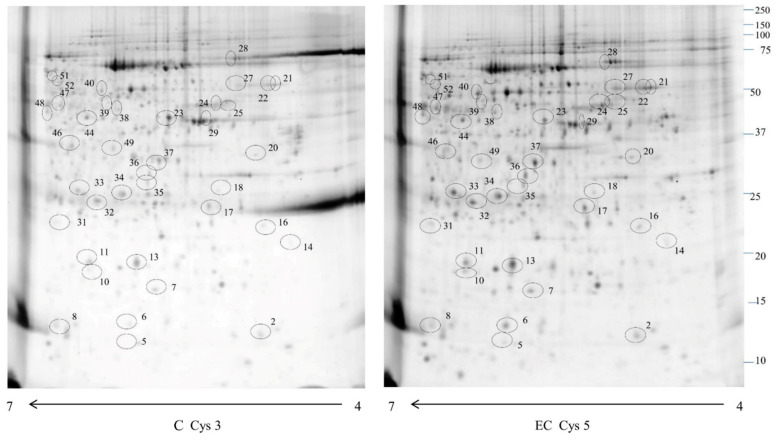
A 2D-DIGE map of normal endometrium (C) and endometrial cancer (EC) phosphoproteome enriched by IMAC columns. Immobilized pH gradient 4–7 strips were used for the first dimension, and 12% polyacrylamide gel for the second dimension. The numbered circles indicate the differentially phosphorylated spots identified in Table 1.

**Figure 2 ijms-24-11987-f002:**
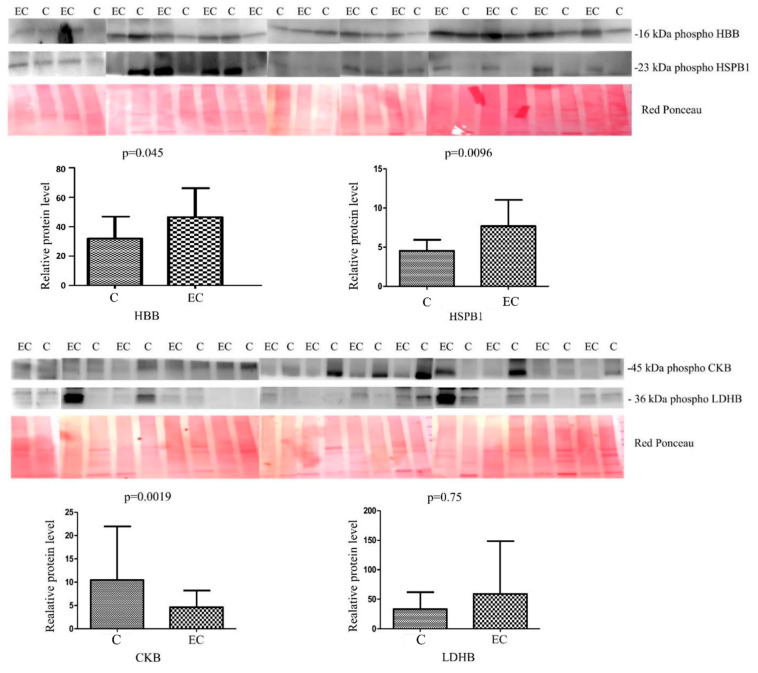
Western blot analysis was utilized to confirm the alteration of phosphorylation of proteins HBB, HSPB1, LDHB and CKB in normal endometrium (C) and endometrial cancer (EC). The intensity of immunostained bands was normalized against the total protein intensities measured from the same blot stained with Red Ponceau. Results are displayed as a histogram (*p* < 0.05), and each bar represents mean ± standard deviation.

**Figure 3 ijms-24-11987-f003:**
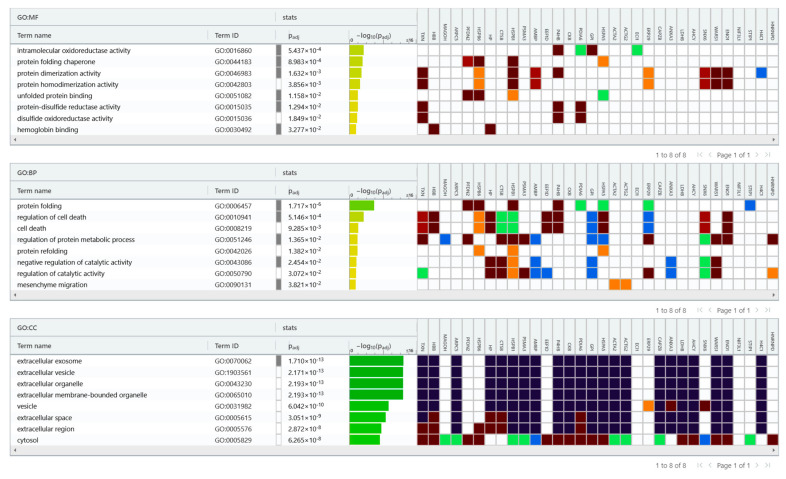
gProfiler classification of the EC phosphoproteins according to their molecular function (MF), biological processes (BP), and cellular component (CC).

**Figure 4 ijms-24-11987-f004:**
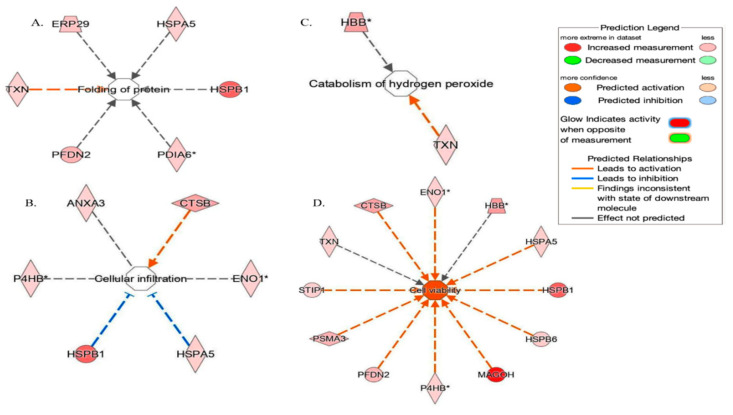
Network build-up from one of the most significant bio-functions: (**A**) Protein folding (Endoplasmic reticulum resident protein 29 (ERP), Endoplasmic reticulum chaperone BiP (HSPA5), Thioredoxin (TNX), Heat shock protein beta-1 (HSPB1), Prefoldin subunit 2 (PFDN2), Protein disulfide-isomerase A6 (PDIA6); (**B**) Cellular infiltration (Annexin A3 (ANXA3), HSPB1, HSPA5, 2-phospho-D-glycerate hydro-lyase (ENO1), Cathepsin B (CTSB), Protein disulfide-isomerase (P4HB); (**C**) Catabolism of hydrogen peroxide Thioredoxin (TNX), Hemoglobin subunit beta (HBB); (**D**) Cellular viability. ENO1, HBB, HSPA5, HSPB1, Heat shock protein beta-6 (HSPB6), Protein mago nashi homolog (MAGOH), P4HB, PFDN2, Proteasome subunit alpha type-3 (PSMA3), Stress-induced-phosphoprotein 1 (STIP), TXN, CTSB. * Isoform indication.

**Table 1 ijms-24-11987-t001:** Different abundance phosphoproteins were identified by mass spectrometry in EC compared to the control tissue.

Spot Number	Accession Number	Protein Description	Gene Symbol	Fold Change	Phosphorylation Sites	*p*-Value
Spot 6	P61326	Protein mago nashi homolog	MAGOH	16.4		0.023
Spot 16	P04792	Heat shock protein beta-1	HSPB1 ^1,2^	10.4	S3 S15 S82 S83	0.045
Spot 37	P07195	L-lactate dehydrogenase B chain	LDHB ^1^	9	S85 S90	0.0059
Spot 8	P68871	Haemoglobin subunit beta	HBB ^1,2^	5.71	T5 S10 S45 T124	0.045
Spot 14	P07858	Cathepsin B	CTSB	5.34		0.013
Spot 17	P25788	Proteasome subunit alpha type-3	PSMA3 ^1,2^	4.4	S9	0.02
Spot 10	Q9UHV9	Prefoldin subunit 2	PFDN2	4.18		0.04
Spot 32	P42126	Enoyl-CoA delta isomerase 1, mitochondrial	ECI1	4.12		0.0062
Spot 31	P63267	Actin, gamma-enteric smooth muscle	ACTG2 ^2^	3.39	S23	0.012
Spot 7	O15511	Actin-related protein 2/3 complex subunit 5	ARPC5 ^1^	3.39	S64	0.01
Spot 34	P30040	Endoplasmic reticulum resident protein 29	ERP29 ^1^	3.33	Y66	0.045
Spot 13	P00738	Haptoglobin	HP	3.2		0.0019
Spot 25	Q15084	Protein disulfide-isomerase A6	PDIA6 ^1,2^	2.82	S427	0.049
Spot 11	O14558	Heat shock protein beta-6	HSPB6 ^1^	2.72	S16	0.0036
Spot 28	P11021	Endoplasmic reticulum chaperone BiP	HSPA5 ^1,2^	2.64	S4 S448 S86 T648	0.0023
Spot 47	A0A2R8Y6G6	2-phospho-D-glycerate hydro-lyase	ENO1 ^1,2^	2.55	S9	0.035
Spot 51	P31948	Stress-induced-phosphoprotein 1	STIP1 ^1^	2.5	S528 S63	0.035
Spot 36	P12429	Annexin A3	ANXA3 ^1^	2.31	S19	0.031
Spot 21	P07237	Protein disulfide-isomerase	P4HB ^1,2^	2.15	S32	0.045
Spot 27	P06744	Glucose-6-phosphate isomerase	GPI ^1^	2.12	S146 S494 S61 T148 T254	0.045
Spot 40	P23381	Tryptophan--tRNA ligase, cytoplasmic	WARS1 ^1^	2.14	S353 S4 S71 T453	0.0063
Spot 20	P29692	Elongation factor 1-delta	EEF1D ^1,2^	1.91	S19 T4 S10	0.039
Spot 35	P47756	F-actin-capping protein subunit beta	CAPZB ^1^	1.86	S211	0.03
Spot 46	Q14103	Heterogeneous nuclear ribonucleoprotein D0	HNRNPD ^1,2^	1.77	S19 T4 S10	0.0038
Spot 39	Q9UNH7	Sorting nexin-6	SNX6 ^1^	1.73	S67 S206	0.045
Spot 38	P23526	Adenosylhomocysteinase	AHCY ^1^	1.68	S68 S70 S80 S84 S85 T82 Y81	0.01
Spot 18	P02760	Protein AMBP	AMBP	1.66		0.035
Spot 33	O75937	DnaJ homolog subfamily C member 8	DNJC8	1.57		0.0045
Spot 2	P10599	Thioredoxin	TXN ^1,2^	1.55	T9	0.045
Spot 49	Q9GZT8	NIF3-like protein 1	NIF3L1	1.52		0.045
Spot 44	P23526	Adenosylhomocysteinase	AHCY ^1^	1.5	S68 S70 S80 S84 S85 T82 Y81	0.037
Spot 29	P62736	Actin, aortic smooth muscle	ACTA2	0.47		0.045
Spot 23	P12277	Creatine kinase B-type	CKB ^1^	0.28	S202 S6 T346 T35	0.0072
Spot 52	P62805	Histone H4	H4C1 ^2^	0.27	S3	0.04

^1^. Phosphoproteins identified in Yongchao Dou et al. [19]. ^2^. Phosphoproteins identified by our LC-MS/MS analysis.

## Data Availability

The data presented in this study are available on request from the corresponding author. The data are not publicly available due to ethical reasons.

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
