# Peer review of "Phospho-DIGE Identified Phosphoproteins Involved in Pathways Related to Tumour Growth in Endometrial Cancer"

_ijms, 2023, doi:10.3390/ijms241511987_

Round 1

Reviewer 1 Report

Thank you for the opportunity to evaluate this interesting paper which sought to evaluate a role of a phosphoproteome and which identify differentially phosphorylated proteins involved in tumorigenic signaling pathways which induce cancer growth. 

Introduction section should be more compressed, exclude the redundant parts. 

Methods and Results of the study are clearly presented.

Discussion section should include limitations of the study. Please provide guidelines for further research in this field. 

Minor editing of English language required.

Author Response

Reviewer 1

Introduction section should be more compressed, exclude the redundant parts.

Our reply: We fixed the introduction.

Discussion section should include limitations of the study. Please provide guidelines for further research in this field.

Our reply: We included the limitation of the study in the discussion and the guidelines for further research in this field.

Minor editing of English language required.

Our reply: We fixed the English language

Reviewer 2 Report

The research investigates the involvement of the phosphoproteome in endometrial cancer tumor growth. Phosphorylation is important for the signal transduction in cancer pathway network.

1. Figures need to be revised to be organized with labels. For instance, the differences between left and right panels are not clear in Figure 1. Figure 2 needs labels in each graph and legends. The detailed information on EC and C of each column may be added.

2. Figure 3 needs more precise explanation of "gProfiler classification" in the legend.

3. Networks with bio-functions may be elaborated with the functions of the centered nodes in Figure 4. Labels of each network may be added. The brief explanation of the phosphoproteins in the networks may be added in the legend.

Please check the references very carefully. 

Some minor proofreading of the manuscript is needed.

Author Response

Reviewer 2

  1. Figures need to be revised to be organized with labels. For instance, the differences between left and right panels are not clear in Figure 1. Figure 2 needs labels in each graph and legends. The detailed information on EC and C of each column may be added.

Our reply: We fixed the figure 1 and 2. We have provided patient information in Supplemental Table 1.

  1. Figure 3 needs more precise explanation of "gProfiler classification" in the legend.

Our reply: We provided more explanation of "gProfiler classification" in Figure 3.

  1. Networks with bio-functions may be elaborated with the functions of the centered nodes in Figure 4. Labels of each network may be added. The brief explanation of the phosphoproteins in the networks may be added in the legend.

Our reply: We fixed the Figure 4 how the reviewer request. We added the explanation of the phosphoproteins in the legend.

Please check the references very carefully.

Our reply: We checked the references.